# In Vivo Effects of Balanced Crystalloid or Gelatine Infusions on Functional Parameters of Coagulation and Fibrinolysis: A Prospective Randomized Crossover Study

**DOI:** 10.3390/jpm12060909

**Published:** 2022-05-31

**Authors:** Agnieszka Wiórek, Piotr K. Mazur, Elżbieta Żurawska, Łukasz J. Krzych

**Affiliations:** 1Department of Anaesthesiology and Intensive Care, Faculty of Medical Sciences in Katowice, Medical University of Silesia, 40-055 Katowice, Poland; lkrzych@sum.edu.pl; 2Department of Cardiovascular Surgery, Mayo Clinic, Rochester, MN 55905, USA; piotr.k.mazur@gmail.com; 3Department of Cardiovascular Surgery and Transplantology, Institute of Cardiology, Medical College, Jagiellonian University, 31-008 Cracow, Poland; 4Central Laboratory, University Clinical Centre of the Medical University of Silesia, 40-055 Katowice, Poland; lab@uck.katowice.pl

**Keywords:** fluid resuscitation, fluid therapy, coagulation and fibrinolysis, rotational thromboelastometry, point-of-care testing, perioperative medicine

## Abstract

Prudent administration of fluids helps restore or maintain hemodynamic stability in the setting of perioperative blood loss. However, fluids may arguably exacerbate the existing coagulopathy. We sought to investigate the influence of balanced crystalloid and synthetic gelatine infusions on coagulation and fibrinolysis in healthy volunteers. This prospective randomized crossover study included 25 males aged 18–30 years. Infusions performed included 20 mL/kg of a balanced crystalloid solution (Optilyte^®^) or 20 mL/kg of gelatine 26.500 Da (Geloplasma^®^) in a random order over a period of 2 weeks. Laboratory analysis included conventional coagulation parameters and rotational thromboelastometry (ROTEM) assays. We confirmed a decrease in fibrinogen concentration and the number of platelets, and prolongation of PT after infusions. Compared to baseline values, differences in the ROTEM assays’ results after infusions signified the decrease in coagulation factors and fibrinogen concentration, causing impaired fibrin polymerization and clot structure. The ROTEM indicator of clot lysis remained unaffected. In the case of both Optilyte^®^ and Geloplasma^®^, the results suggested relevant dilution. Gelatine disrupted the process of clot formation more than balanced crystalloid. Infusions of both crystalloid and saline-free colloid solutions causing up to 30% blood dilution cause significant dilution of the coagulation factors, platelets, and fibrinogen. However, balanced crystalloid infusion provides less infusion-induced coagulopathy compared to gelatine.

## 1. Introduction

Commercially available crystalloid intravenous (IV) fluids constitute a heterogeneous group of solutions designed as a non-physiologic source of free water, glucose, and electrolytes. They provide volume expansion and are frequently used to correct and stabilize the electrolyte balance, depending on the composition of the solution. Synthetic colloids are used as typical volume expanders. Therapy with either crystalloids or colloids is useful for replenishing extravascular fluid deficits and intravascular hypovolaemia [1]. Prudent administration of such fluids helps restore or maintain hemodynamic stability in the setting of perioperative blood loss [1]. The question of which IV fluid is superior to the others in the perioperative setting remains yet to be answered, and the debate on the best infusion strategy is ongoing [2].

Personalized goal-directed volume therapy, based on complex patient monitoring, seems to be beneficial in reducing the risk of complications [3,4]. However, intravenous fluids may arguably exacerbate the existing coagulopathy due to dilution, especially if they are deprived of coagulation factors and blood components [5]. Dilutional coagulopathy and the specific impact of fluids on clot formation and its stability are of utmost importance in the setting of perioperative blood loss [6]. Monitoring of this interesting process has been previously performed using conventional or viscoelastic coagulation tests [7,8].

Most studies concerning the in vivo effects of infused solutions on coagulopathy were based on 0.9% sodium chloride solution or hydroxyethyl starches (HES), which are currently used less frequently due to known adverse effects [9]. Normal saline solution (0.9% NaCl; NS) infused in large volumes needed for fluid resuscitation leads to hyperchloremic acidosis [10]. HES solutions were reported to cause kidney injury and increased the need for dialysis in two large-scale randomized controlled trials [11,12]. In vivo studies comparing the effects of NS vs. Ringer’s lactate [13], 6% HES vs. Ringer’s lactate [14], and HES vs. albumin [15] provide conclusions, which are hardly applicable to modern clinical practice. An in vivo study on 60 out of 240 planned patients by Pfortmueller et al. comparing NS with an acetate-buffered balanced crystalloid solution was terminated prematurely due to excess hyperchloremic acidosis in the NS group [16].

The in vitro effects of balanced crystalloid and balanced colloid infusions on coagulation and fibrinolysis were previously investigated by our group [17]. We indicated that for minimum interference with coagulation and fibrinolysis, fluid resuscitation with balanced crystalloid solutions causing the 20% blood dilution seemed safe. When further volume expansion is required, gelatines appeared a reasonable choice for second-line treatment. HES should be avoided due to its deleterious effect on hemostasis.

In the current report, we sought to investigate in the in vivo setting the influence of different fluid infusions used in modern clinical practice (i.e., balanced crystalloids and synthetic gelatines) on coagulation, using frequently applied perioperatively standard laboratory tests and rotational thromboelastometry (ROTEM) in a randomized study on healthy volunteers.

## 2. Materials and Methods

### 2.1. Study Participants

This prospective randomized crossover study was designed to include healthy male volunteers aged 18–30 years in the American Society of Anaesthesiologists Physical Status class I. All participants were recruited by informational flyers distributed among the hospital and academic staff members. The exclusion criteria were female sex, blood type O, a positive history of any acute diseases in the last four weeks, chronic diseases, any diagnosed haemostatic disorders, history of anticoagulation, known bleeding diathesis, and any pharmacotherapy in the previous week. The blood type of the participants was determined through laboratory authorized blood type test results shown by the candidate. Participants were informed about the prohibition of alcohol intake, excessive exercise, and stress on the day before blood sampling. There was no reimbursement of costs or financial support for participating in the project. The patient flow diagram is shown in Figure 1. All demographic and medical data were recorded prospectively.

The study was approved by the Ethics Committee of the Medical University of Silesia in Katowice, Poland (KNW/0022/KB1/159/II/15/16/18/19). Written informed consent was obtained from all participants. The study was registered online in the ClinicalTrials.gov database (NCT05148650). The CONSORT Statement (Consolidated Standards of Reporting Trials) was applied for appropriate data reporting.

### 2.2. Study Design and Interventions

Infusions performed included 20 mL/kg of a balanced crystalloid solution (Optilyte^®^, Fresenius Kabi, Bad Homburg, Germany) or 20 mL/kg of gelatine 26.500 Da (Geloplasma^®^, Fresenius Kabi, Bad Homburg, Germany) in a random order over a period of 14 days (washout period). The infusions were performed through an 18G intravenous cannula inserted into an antecubital vein on the non-dominant limb at a rate of 1000 mL/h. Based on this infusion rate and depending on the amount of fluid to be infused, calculated from the participant’s weight, the infusion took between 90 to 120 min. The first blood sample was collected straight after insertion of the IV cannula, before the start of the infusion, and the second immediately after the infusion of the test solution, from a separate venipuncture on a contralateral extremity.

### 2.3. Randomization

Enrolled participants were randomly assigned using a 1:1 ratio to receive either the Optilyte^®^ or Geloplasma^®^ infusion. Before the onset of the study, opaque envelopes containing equalized numbers of cards indicating the type of solution to be infused (Optilyte^®^, *n* = 13 or Geloplasma^®^, *n* = 12) were sealed, shuffled, and numbered from 1 to 25. The principal investigator (AW) enrolled all participants. Consecutive participants received numbers corresponding to consecutive numbers on envelopes. The investigator responsible for administering infusions opened the envelopes just before entering the lab and was not blinded to the test solutions. All participants received the infusions based on the same principle of dose calculation and infusion rate. Participants and the Central Laboratory team performing the standard laboratory tests were blinded to test solution assignment. The investigator responsible for running the ROTEM assays was also blinded to the type of infused fluid. After the washout period of 2 weeks (14 days), the participants returned to receive the second type of fluid in the exact same way and volume as calculated for the first infusion. 

### 2.4. Laboratory Investigations

Blood samples were collected from an antecubital vein with minimal stasis at 3 to 5 PM on the day appointed with the eligible participating volunteer after confirming the participant’s abidance of the study conditions concerning the prohibition of alcohol intake, excessive exercise, and stress on the day before blood sampling. Two blood samples, 12.5 mL each, were taken just before and immediately after the fluid infusion, using a vacuum system (BD Vacutainer^®^ Warsaw, Poland); a total of 25 mL of blood was collected. The blood sample after the fluid infusion was collected through an IV cannula from a separate venipuncture on a contralateral extremity to avoid interference with the clear fluid residue in the first intravenous cannula. In both cases, during pre-dilution and post-dilution blood sampling, the first 5 mL of blood were disposed of due to the possible interference with vascular stasis on the measurement results. Next, blood was collected through a vacuum system into the test tubes corresponding to the laboratory analysis for full blood count (containing ethylenediaminetetraacetic acid (EDTA), coagulology, and D-dimer concentration (containing 3.2% buffered sodium citrate), and ROTEM assays (containing 3.2% buffered sodium citrate).

Standard laboratory tests for determining coagulation status were performed, including fibrinogen concentration, D-dimer concentration, activated partial thromboplastin time (APTT), prothrombin time (PT), with the calculation of international normalized ratio (INR), hematocrit, hemoglobin concentration, platelet count (PLT), mean platelet volume (MPV), platelet distribution width (PDW), and platelet-large cell ratio (P-LCR). The fibrinogen concentration was assessed based on the Clauss method for the quantitative determination of fibrinogen, using thrombin to measure fibrinogen in human citrated plasma on the IL Coagulation Systems [18]. The reference range values for the investigated parameters of standard laboratory tests and their short descriptions are listed in Appendix A.

Rotational thromboelastometry coagulation analysis was carried out using a ROTEM delta analyzer (Tem Innovations GmbH, Munich, Germany), following the manufacturer’s instructions. The assays were allowed to run for 60 min. Assays were run immediately after blood sampling to minimize a preanalytical error, as the ROTEM analyzer was checked and prepped directly before the admission of the participant and available in the same room where the infusions took place. Three ROTEM assays were run simultaneously, INTEM, EXTEM, and FIBTEM. All ROTEM analyses were performed by the same investigator. The parameters of interest measured in the three assays were clotting time (CT), clot formation time (CFT), alpha angle (AA), the amplitude at different time points (minutes) (A10, A20), maximum clot firmness (MCF), and maximum lysis (ML). The maximum clot elasticity (MCE) for EXTEM and FIBTEM assays was calculated with the following formula: MCE = 100 × MCF/100–MCF. Assessment of platelet contribution to clot strength was measured according to the formula ΔMCE = MCEEXTEM − MCEFsIBTEM. The reference range values for the selected ROTEM parameters and their short descriptions are depicted in Appendix A.

No changes (protocol violations) were required to the study protocol after the initiation of the study. The summarized graphical illustration of the study protocol is depicted in Figure 2.

Only completed cases (*n* = 25) were included in the final study analysis; we experienced no losses of participants and no cases of protocol breaking or deviation through the course of the study. We opted for a crossover study design in which each participant undergoes both interventions, which differed only in the type of fluid infused and were otherwise identical in nature. Participants had two weeks of a washout period between crossovers and returned on the 14th day from completing the first arm of the protocol for their second visit to complete the study. Volunteers completed both arms of the study within 14 days. The primary overall clinical endpoint was coagulation and fibrinolysis impairment after infusion of balanced crystalloid and balanced colloid. The secondary clinical outcome assessed was safety as the potential of adverse events after fluid infusion.

### 2.5. Statistical Analysis

Statistical analysis was performed using MedCalc v.18 software (MedCalc Software, Ostend, Belgium). No a priori power or sample calculation was performed. Quantitative variables were depicted using medians and interquartile ranges (IQR). The Shapiro–Wilk test was used to verify their distributions. Qualitative variables were described using frequencies and percentages. The Wilcoxon signed-rank test for paired samples was utilized for determining between-group differences for pre- and post-dilution results for each fluid individually, as well as for the differences between pre-dilution results compiled for both types of fluids and post-dilution results comparing crystalloid and colloid. The U-Mann–Whitney test was utilized for determining between-group differences in the volume of administered fluid. All tests were two-sided. A *p*-value of <0.05 was considered significant.

## 3. Results

Out of 40 individuals screened for eligibility between February 2021 and May 2021, 25 healthy Caucasian males were included. The median age of participants was 25 years (IQR, 23–29), median weight 80 kg (71–85), and median height 181 cm (176–184). The most frequent blood type was A Rh-positive (*n* = 13/52%), followed by B Rh-positive (*n* = 8/32%), A Rh-negative (*n* = 2/8%), and B Rh-negative (*n* = 2/8%).

The baseline characteristics and differences between the studied parameters in the study population before the infusion of Optilyte^®^ and before the infusion of Geloplasma^®^ are depicted in Table 1 and Table 2.

Based on the volume of fluid to be infused calculated for each participant’s body weight, the median volume of infused fluids was 1600 mL (IQR, 1430–1700 mL), and it was the same for the crystalloid and colloid.

### Effects of Fluids Infusion on Coagulation

The effects of both investigated fluids on coagulation and the differences between the paired measurements are presented in Table 3 for standard laboratory tests and Table 4 for ROTEM assays. 

The comparative analysis in the standard laboratory tests confirmed differences in fibrinogen concentration, platelet count, and PT before and after infusion (Table 3).

Both in the case of Optilyte^®^ and Geloplasma^®^, the results suggested dilution. Median haematocrit drop after Optilyte infusion was 7% (IQR 6–10, min 0, max 15), and 17% (IQR 15–20, min 12, max 27) after Geloplasma infusion (*p* < 0.0001). They were more pronounced after gelatine infusion based on the PT and INR after gelatine being longer than PT and INR after crystalloid, and the fibrinogen concentration and platelet count values after gelatine being lower than the same parameters values after crystalloid in the between-group differences compared after both infusions. The fibrinolysis-dependent variable (D-dimers) was unaffected by either of the fluids. 

There were differences in most studied parameters in the EXTEM and INTEM assays before and after infusion, both in the case of Optilyte^®^ and Geloplasma^®^. Differences between the FIBTEM parameters before and after the infusions of both test solutions signify the decrease in fibrinogen concentration, causing impaired fibrin polymerization and impaired clot structure. A comparison of the differences between results acquired in the post-dilution samples for both solutions shows that although both fluids caused relevant dilution, gelatine disrupted the process of clot formation more than balanced crystalloid, as seen in the more distinct prolongation of time needed for the clot to form (CFT), and the more pronounced decrease in the kinetics of clot aggregation (AA) after gelatine. The same can be reported for more diminished clot strength after gelatine, with more reduced amplitude of clot firmness in given time points (A10, A20) as well as more decreased maximum clot firmness than after crystalloid. Platelet-dependent variable (ΔMCE) was also disrupted more after the gelatine infusion (Table 4). 

No harm or adverse side effects were recorded throughout the study period.

## 4. Discussion

This study is the first among in vivo studies to show that balanced crystalloid and synthetic saline-free colloid infusions may cause acute changes in coagulation, as demonstrated using standard laboratory tests and ROTEM. The effects of such infusions on clot formation could become clinically relevant, particularly in the perioperative setting or emergency departments of volume depletion during acute blood loss.

Some previous studies identified FIBTEM as the best predictive ROTEM assay for recognizing dilutive coagulopathy [15]. MCF parameter in FIBTEM assay may be helpful in guiding fibrinogen replacement therapy during perioperative management of hypofibrinogenemic patients, as it requires a specific assessment of the patient’s overall hemostasis, evaluating both the bleeding and thrombotic risk [19]. This assay allows for the differentiation between deficiency of platelets or fibrinogen while remaining dependent on coagulation factors concentration. Fibrinogen is one of the first coagulation factors to reach critically low levels during haemodilution, and the first coagulation factor to be depleted during significant bleeding, as seen in previous in vitro studies [20,21]. Current research recognizes that clinical manifestations of bleeding are dependent on the fibrinogen concentrations, with the “bleeding phenotype” usually developing in patients with fibrinogen concentration lower than 1.0 g/L, however, on the other hand, when a mean activity level of fibrinogen is at least 0.7 g/L, it prevents spontaneous haemorrhaging [22]. A decrease in fibrinogen concentration impairs the goal of the coagulation cascade, which is creating an insoluble fibrin polymer, ensuring structural stability, strength, and adhesive surfaces to growing blood clots [23]. No effects on fibrinolysis were recorded for both solutions. In general, there is a measurable tendency for both balanced crystalloid and colloid to cause ROTEM-detectable coagulopathies even at low levels of dilution, such as in our study [24]. However, as reflected by the differences in coagulation studies acquired after both infusions, the gelatine solution diluted the coagulation factors, platelets, and fibrinogen to a greater extent, contributing to the worsening of clot formation and strength, disruption of fibrin polymerization, and platelet aggregation. 

It should be taken into consideration that modern resuscitation practices for bleeding management consider “haemostatic resuscitation” (HR) as the leading strategy, with the delivery of blood, blood components (plasma, platelets, and fibrinogen), and haemostatic agents, while surgical control of bleeding is achieved [25]. Nevertheless, based on our observations, it seems that in the event of a delay in providing HR and achieving surgical control, balanced crystalloids may be a better choice than colloids for replenishing lost intravascular volume. Even more so, since the supposedly superior volume effect of gelatines is repeatedly undermined [26,27].

Even though our results acquired from standard laboratory tests such as the PT, INR, platelet count, and fibrinogen concentration seem to very well reflect the state of dilutional consequences on their own, conditioning the clinical decisions solely on their basis is not devoid of shortages [28]. Standard laboratory tests can indicate prolongation of the time needed for the clot to form, the influence of anticoagulant drugs, and, through the D-dimer concentration, can inform about the active process of fibrinolysis due to the overactivation of the coagulation system [28]. However, parameters derived from the standard laboratory tests and platelet counts are measured from patient plasma and only provide static numbers with no information regarding functionality [15,29]. Moreover, the time needed to get the results of the standard laboratory tests from the hospital laboratory can be as long as 45-60 min, even if ordered as urgent. Viscoelastic point-of-care tests, on the other hand, such as rotational thromboelastometry (ROTEM), provide the results as quickly as minutes upon blood collection. That is why in a situation of a patient’s rapidly deteriorating clinical state requiring fast and decisive proceedings, viscoelastic point-of-care testing such as ROTEM allows for a safer, more accurate, personalized, goal-directed bleeding management aimed at replenishing precisely what the patient lacks [30].

Studies aiming to evaluate the differences in colloids’ and crystalloids’ influence on coagulation are often based on in vitro dilution of collected blood samples [31,32]. Results of such a strategy are shown, among others, in the studies by Winstedt et al., Getrajdman et al., and Czempik et al. However, each of these studies approached the issue differently with their selection of study population, the extent of haemodilution, and tested fluids. These studies showed that parameters reflecting clot formation and clot strength were impaired by haemodilution, and the effect was more enhanced by the colloid than crystalloid, although sometimes the effect was only distinct with a large degree of more than 40% of dilution. [17,20,33]. Our results showed that even though the difference in ROTEM parameters before and after infusion suggested the potential of dilutional disturbances for both fluids, gelatine solution induced stronger effects even with a dilution ratio up to much less than 40%.

The issue of fluids’ volume effect variety and its influence on coagulation has for a long time been a focal point of an extensive research area such as perioperative medicine. In the in vivo conditions, the volume effect of different colloids has been studied, for instance, among patients scheduled for orthopaedic surgery, considering the diluted blood samples collected at baseline before induction, before surgical incision, and every 90 min after that [34]. One of the first studies to be conducted in vivo to investigate the influence of combined crystalloid/colloid intravenous administration on the coagulation system also focused on the orthopaedic patients during primary knee replacement surgery, some of whom were suffering from comorbidities placing them in the ASA-PS III class [35,36]. Even earlier in vivo studies focused particularly on the effect of gelatine infusion on coagulation in a group of six healthy male participants and reported a significant impairment of primary haemostasis and thrombin generation resulting in an almost 2-fold increase in bleeding time at 60 and 120 min compared with 0.9% NaCl [37]. The aforementioned studies reported longer maintenance of the intravascular volume effect of colloids than crystalloids and, as a result, a more significant influence of the colloids (starches more than gelatine) on the speed and quality of the clot formation [34,35]. However, those studies were not deprived of confounding factors that could have influenced the results. Patients participating in them were often burdened with severe comorbidities that could have influenced the metabolism of the administered drugs and fluids and the overall condition of the vascular endothelium affecting its permeability, as observed in time-dependent repeated measurements [34,36]. Furthermore, even the administration of anaesthesia itself may be considered a confounding factor through the individual effects of various intravenous and volatile anaesthetics on vascular permeability [38]. To eliminate potential confounders, our study was conducted in vivo amongst healthy participants, and the blood samples were collected immediately upon the end of iatrogenically induced haemodilution. This minimized the time for the occurrence of any varied effect of fluid metabolism, which can vary individually even in healthy people. We also did not focus on further follow-up after later time points, as the study group comprised subjects with no suspected risk of kidney injury and delayed filtration. Even so, this should provide more reliable results, as we eliminated some factors creating limitations for in vitro coagulation assessment. Limitations are emerging for example from shear stress applied with viscoelastic haemostatic assays being substantially lower than those found in the human circulation, the acid-base balance disturbances, and the fact that decreased haematocrit may increase fibrin thread formation in the reaction chamber [39,40]. In vitro coagulation assessment also poses an increased risk of the in vitro blood samples deterioration before analysis due to, for example, storing at suboptimal temperature, which may lead to predominant prolongation of the initiation phase of clot formation and inhibition of fibrinogen synthesis [41].

Multiple factors, including the high cost of blood components, limited resource availability, and associations of increased morbidity, the incidence of nosocomial infections, multiple organ failure, lung injury, cardiac overload, and increased mortality with allogenic blood product infusions, contribute to the use of crystalloids and colloids when clinical signs of bleeding appear [42]. The examples of justified utilization of synthetic fluid to minimize infusions also include, e.g., volume priming in the extracorporeal circulation circuits (ECC). Haemodilution in ECC is advantageous to a certain degree due to lowering blood viscosity and the potential of improving microcirculatory perfusion [21]. However, for the benefits to be preserved and to prevent the increased morbidity associated with low haematocrit values, the minimally invasive extracorporeal circulation circuits, requiring small priming volumes of about 600–700 mL, are now preferred and show a reduced risk of red blood cells transfusion [21]. The use of functional point-of-care tests also supports the rational disposition of blood products, and thromboelastometry, thromboelastography, alongside activated clotting time are among the most commonly utilized POCTs in the operating room environment. [43,44,45].

### Study Limitations

The studied cohort was small and no a priori sample size calculation was performed. We based the study group size on the available published literature with a similar study methodology [17]. Furthermore, the findings in healthy volunteers may differ from those of the patients treated in the emergency unit or operating room due to massive bleeding, hypothermia, and coagulopathy. We also did not include follow-up data covering the time-dependent evolution of detected coagulation impairment, due to selection criteria adopted for our study group. Our study enrolment was based on the declaration of no known comorbidities, including the minimal risk of any kidney insufficiency that could potentially cause prolonged persistence of haemodilution effect on coagulation. However, the study group was constructed to avoid any potential confounding variables that could otherwise influence the differences in parameters caused solely by intravenous fluid administration, such as the “lethal diamond” of hypothermia, acidosis, coagulopathy, and hypocalcemia significant in trauma-induced coagulopathy and cardiopulmonary bypass-induced coagulopathy [41,46]. Thirdly, the degree of induced dilution had to remain safe for the participants. Therefore, it may have been insufficient to report further coagulation abnormalities even with the ROTEM analysis. ROTEM results might not always correlate with clinical signs of bleeding, as there is no blood flow or interactions with endothelium that affect coagulation in vivo. Finally, a gender criterion was implemented for patients’ exclusion, which might impact the external validity of the results in the female population. Due to possible additional blood loss associated with menstruation and the proven influence of hormonal changes on the coagulation process, women were excluded. Patients with blood type O were also excluded, as they may have genetically lower plasma von Willebrand factor levels than those with non-O blood, increasing their risk of haemorrhage, which could act as a confounding variable in the study environment, although it is not limiting the clinical use of ROTEM for type O blood [47].

## 5. Conclusions

Infusions of both balanced crystalloid and saline-free colloid solutions causing up to 30% blood dilution cause significant dilution of the coagulation factors, platelets, and fibrinogen. However, balanced crystalloid infusion causes less infusion-induced coagulopathy compared to gelatine.

## Figures and Tables

**Figure 1 jpm-12-00909-f001:**
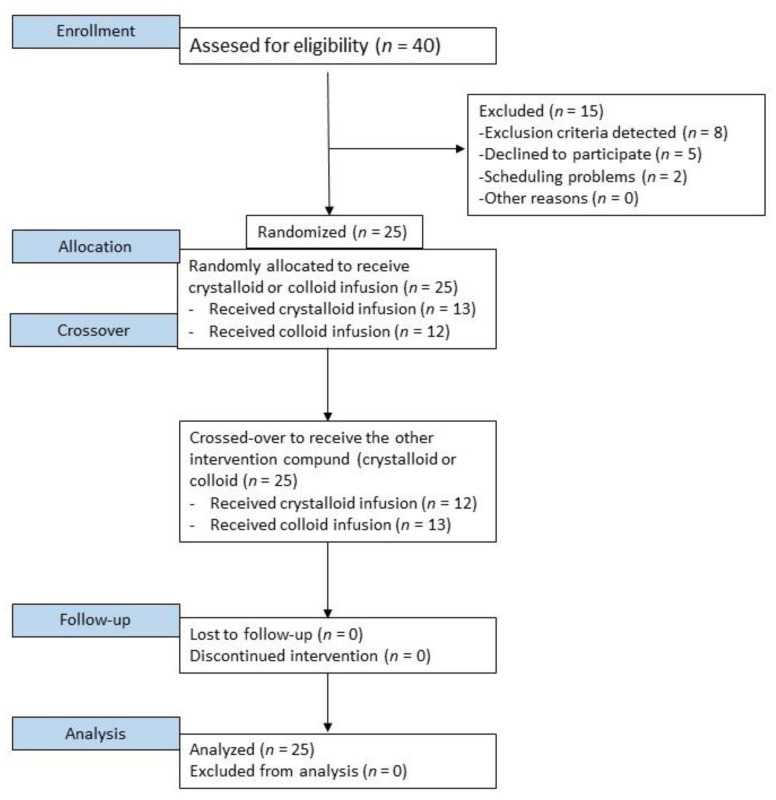
Study flow chart.

**Figure 2 jpm-12-00909-f002:**
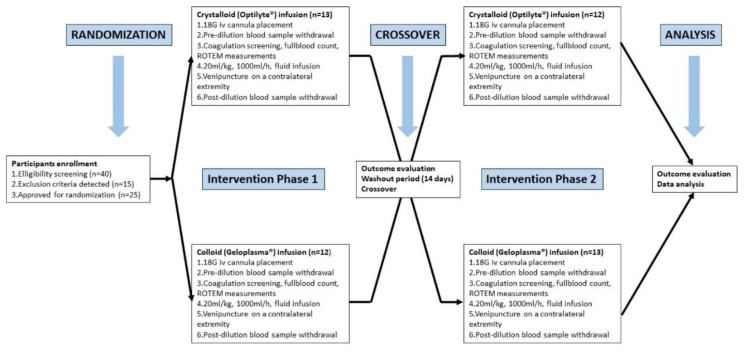
Study protocol.

**Table 1 jpm-12-00909-t001:** Baseline characteristics and differences between the studied parameters in the study population before the infusion of Optilyte^®^ and before the infusion of Geloplasma^®^ for standard laboratory tests ^1^.

Parameter	Median (IQR) before Optilyte^®^ (*n* = 25)	Median (IQR) before Geloplasma^®^ (*n* = 25)	‘*p*’
APTT (s)	29.7 (28.2–33.0)	30.8 (28.7–33.0)	0.71
PT (s)	11.7 (11.4–12.0)	11.7 (11.4–12.5)	0.36
INR	1.01 (0.98-1.04)	1.01 (0.95–1.08)	0.42
Fibrinogen (mg/dL)	198 (183–243)	200 (177–246)	0.99
D-dimer (ug/mL)	171 (109–303)	198 (133–222)	0.62
PLT (10^3^/µL)	244 (226–280)	238 (222–262)	0.16
PDW (fl)	11.6 (11.0–12.9)	12.0 (10.9–13.3)	0.73
MPV (fl)	10.2 (9.9–10.8)	10.1 (9.8–11.0)	0.15
P-LCR (%)	26.8 (24.1–31.3)	27.5 (23.0–33.4)	0.19

^1^ Values are medians and interquartile ranges (IQR). Abbreviations: APTT, activated partial thromboplastin time; D-dimer concentration; Fibrinogen, fibrinogen concentration; INR, international normalized ratio; MPV, mean platelet volume; PDW, platelet distribution width; P-LCR, platelet-large cell ratio; PLT, platelet count; PT, prothrombin time; *p*, differences between pre-dilution results for both types of fluid.

**Table 2 jpm-12-00909-t002:** Baseline characteristics and differences between the studied parameters in the study population before the infusion of Optilyte^®^ and before the infusion of Geloplasma^®^ for ROTEM parameters ^1^.

Assay	Parameter	Median (IQR) before Optilyte^®^ (*n* = 25)	Median (IQR) before Geloplasma^®^ (*n* = 25)	‘*p*’
EXTEM	CT (s)	61 (58–66)	60 (59–66)	0.86
	CFT (s)	131 (118–148)	131 (118–147)	0.30
	AA (◦)	65 (62–68)	65 (63–68)	0.52
	A10 (mm)	51 (49–54)	51 (49–55)	0.84
	A20 (mm)	58 (56–61)	59 (56–61)	0.60
	MCF (mm)	59 (57–62)	59 (57–61)	0.24
	ML (%)	9 (7–11)	9 (7–10)	0.77
	MCE (dynes/cm^2^)	144 (133–165)	144 (131–158)	0.18
	ΔMCE	138 (125–155)	135 (122–150)	0.14
INTEM	CT (s)	166 (161–177)	170 (167–178)	0.64
	CFT (s)	75 (68–83)	75 (66–88)	0.62
	AA (◦)	75 (74–76)	75 (73–77)	0.98
	A10 (mm)	56 (53–60)	57 (53–58)	0.59
	A20 (mm)	61 (59–65)	61 (59–64)	0.57
	MCF (mm)	62 (59–65)	62 (59–65)	0.55
	ML (%)	7 (5–8)	7 (5–9)	0.45
	MCE (dynes/cm^2^)	163 (144–188)	163 (144–186)	0.78
FIBTEM	CT (s)	61 (57–68)	64 (60–67)	0.49
	A10 (mm)	8 (7–10)	8 (7–11)	0.13
	A20 (mm)	9 (8–11)	8 (7–12)	0.18
	MCF (mm)	9 (8–11)	8 (7–12)	0.30
	MCE (dynes/cm^2^)	10 (8–13)	9 (8–14)	0.25

^1^ Values are medians and interquartile ranges (IQR). Abbreviations: A10, clot firmness amplitude measured after 10 min; A20, clot firmness amplitude measured after 20 minutes; AA, alpha angle; CFT, clot-forming time; CT, clotting time; MCE, maximum clot elasticity; MCF, maximum clot firmness; ML, maximum lysis; *p*, differences between pre-dilution results for both types of fluid.

**Table 3 jpm-12-00909-t003:** Effect of test solutions on parameters in standard laboratory tests of coagulation and fibrinolysis ^1^.

Parameter	Crystalloid (Optilyte^®^) (*n* = 25)			Colloid (Geloplasma^®^) (*n* = 25)			Post-Dilution Between-Group Differences
	Before	After	‘*p* #’	Before	After	‘*p* *’	‘*p* †’
APTT (s)	29.7 (28.2–33.0)	30.8 (28.7–33.8)	0.25	30.8 (28.7–33.0)	31.7 (29.7–34.0)	0.06	0.22
PT (s)	11.7 (11.4–12.0)	12.0 (11.6–12.6)	<0.0001	11.7 (11.4–12.5)	12.7 (12.2–13.6)	<0.0001	0.0001
INR	1.01(0.98–1.04)	1.03 (1.00–1.08)	<0.0001	1.01 (0.95–1.08)	1.09 (1.05–1.17)	<0.0001	0.0001
Fibrinogen (mg/dL)	198 (183–243)	173 (164–214)	<0.0001	200 (177–246)	157 (138–187)	<0.0001	<0.0001
D-dimer (ug/mL)	171 (109–303)	182 (112–296)	0.44	198 (133–222)	184 (150–378)	0.15	0.20
PLT (10^3^/µL)	244 (226–280)	216 (193–238)	<0.0001	238 (222–262)	185 (168–201)	<0.0001	<0.0001
PDW (fl)	11.6 (11.0–12.9)	11.7 (10.6–12.7)	0.23	12.0 (10.9–13.3)	11.7 (10.9–13.4)	0.41	0.12
MPV (fl)	10.2 (9.9–10.8)	10.4 (9.8–10.7)	0.67	10.1 (9.8–11.0)	10.3 (10.0–11.0)	0.39	0.06
P-LCR (%)	26.8 (24.1–31.3)	28.0 (23.4–30.3)	0.97	27.5 (23.0–33.4)	27.6 (24.4–33.2)	0.84	0.12

^1^ Values are medians and interquartile ranges (IQR). Abbreviations: APTT, activated partial thromboplastin time; D-dimer concentration; Fibrinogen, fibrinogen concentration; INR, international normalized ratio; MPV, mean platelet volume; PDW, platelet distribution width; P-LCR, platelet-large cell ratio; PLT, platelet count; PT, prothrombin time; *p* #, differences between pre-dilution and post-dilution results for balanced crystalloid; *p* *, differences between pre-dilution and post-dilution results for gelatine solution; *p* †, differences between post-dilution results for both types of fluid.

**Table 4 jpm-12-00909-t004:** Effect of test solutions on parameters of rotational thromboelastometry, in INTEM, FIBTEM, and EXTEM ^1^.

Assay	Parameter	Crystalloid (Optilyte^®^) (*n* = 25)			Colloid (Geloplasma^®^) (*n* = 25)			Post-Dilution Between-Group Differences
		Before	After	‘*p* #’	Before	After	‘*p* *’	‘*p* †’
EXTEM	CT (s)	61 (58–66)	60 (57–65)	0.98	60 (59–66)	61 (58–65)	0.59	0.88
	CFT (s)	131 (118–148)	135 (120–155)	0.02	131 (118–147)	149 (128–172)	<0.0001	0.0006
	AA (◦)	65 (62–68)	64 (61–67)	0.14	65 (63–68)	62 (58–65)	<0.0001	0.0025
	A10 (mm)	51 (49–54)	50 (47–53)	0.002	51 (49–55)	48 (44–51)	<0.0001	0.0047
	A20 (mm)	58 (56–61)	57 (55–60)	0.003	59 (56–61)	56 (52–58)	<0.0001	0.015
	MCF (mm)	59 (57–62)	58 (56–62)	0.007	59 (57–61)	56 (53–60)	<0.0001	0.016
	ML (%)	9 (7–11)	8 (8–11)	0.86	9 (7–10)	8 (6–11)	0.23	0.54
	MCE (dynes/cm^2^)	144 (133–165)	138 (126–163)	0.004	144 (131–158)	127 (113–152)	0.0001	0.01
	ΔMCE	138 (125–155)	131 (118–152)	0.013	135 (122–150)	119 (106–144)	0.0001	0.015
INTEM	CT (s)	166 (161–177)	162 (153–175)	0.015	170 (167–178)	169 (150–180)	0.027	0.90
	CFT (s)	75 (68–83)	81 (69–90)	0.25	75 (66–88)	98 (88–113)	<0.0001	<0.0001
	AA (◦)	75 (74–76)	74 (73–76)	0.18	75 (73–77)	71 (68–73)	<0.0001	<0.0001
	A10 (mm)	56 (53–60)	55 (52–58)	0.021	57 (53–58)	50 (47–52)	<0.0001	<0.0001
	A20 (mm)	61 (59–65)	61 (58–63)	0.032	61 (59–64)	56 (53–58)	<0.0001	<0.0001
	MCF (mm)	62 (59–65)	61 (58–64)	0.014	62 (59–65)	56 (54–59)	<0.0001	<0.0001
	ML (%)	7 (5–8)	7 (5–8)	0.28	7 (5–9)	7 (5–9)	0.95	0.045
	MCE (dynes/cm^2^)	163 (144–188)	156 (138–176)	0.008	163 (144–186)	127 (116–144)	<0.0001	<0.0001
FIBTEM	CT (s)	61 (57–68)	64 (59–67)	0.27	64 (60–67)	64 (60–71)	0.34	0.62
	A10 (mm)	8 (7–10)	7 (6–9)	0.016	8 (7–11)	6 (5–8)	<0.0001	0.003
	A20 (mm)	9(8–11)	8 (7–11)	0.024	8 (7–12)	7 (6–9)	<0.0001	0.0003
	MCF (mm)	9 (8–11)	8 (7–10)	0.011	8 (7–12)	7 (6–8)	<0.0001	0.0008
	MCE (dynes/cm^2^)	10 (8–13)	9 (8–11)	0.005	9 (8–14)	8 (6–9)	<0.0001	0.0005

^1^ Values are medians and interquartile ranges (IQR). Abbreviations: A10, clot firmness amplitude measured after 10 min; A20, clot firmness amplitude measured after 20 min; AA, alpha angle; CFT, clot-forming time; CT, clotting time; MCE, maximum clot elasticity; MCF, maximum clot firmness; ML, maximum lysis; *p* #, differences between pre-dilution and post-dilution results for balanced crystalloid; *p* *, differences between pre-dilution and post-dilution results for gelatine solution; *p* †, differences between post-dilution results for both types of fluid.

## Data Availability

The datasets used and/or analyzed during the current study are available from the corresponding author on reasonable request.

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
