# Peer review of "In Vivo Effects of Balanced Crystalloid or Gelatine Infusions on Functional Parameters of Coagulation and Fibrinolysis: A Prospective Randomized Crossover Study"

_jpm, 2022, doi:10.3390/jpm12060909_

Round 1

Reviewer 1 Report

In the current study, authors sought to investigate in the in vivo setting the influence of balanced crystalloids and synthetic gelatines on coagulation, using frequently applied perioperatively standard laboratory tests and rotational thromboelastometry (ROTEM) in a randomized study on healthy volunteers. They found that infusions of both solutions that caused blood dilution of up to 30% resulted in significant dilution of clotting factors, platelets, and fibrinogen. However, they showed that balanced infusion of crystalloids caused less infusion-induced coagulopathy than gelatin. 

The paper is well written and provides helpful information for the readers. Therefore, this paper has the potential to be accepted.

Author Response

Reviewer #1

We would like to kindly thank you for your extensive review, substantive approach, and valuable comments and observations concerning the quality of our study. Thank you for your approval of its overall value. Thank you also for your comments and observations acknowledging the potential of our research.

Reviewer 2 Report

The authors compared the in vivo effect of gelatin and crystalloid and gelatin and came to the result, that gelatin impairs fibrin polymerisation more than crystlloids.

Comments:

The effect of colloids, at least gelatins, have been described in the past very sufficiently also under clinical conditions.

Mittermayr M, et al. Hemostatic changes after crystalloid or colloid fluid administration during major orthopedic surgery: the role of fibrinogen administration. Anesth Anlag 2007 Oct;105(4):905-17. 

Innerhofer P et al. The effects of perioperatively administered colloids and crystalloids on primary platelet-mediated hemostasis and clot formation. Anesth Analg. 2002 Oct;95(4):858-65.

Fries D et al. The effects of perioperatively administered crystalloids and colloids on concentrations of molecular markers of activated coagulation and fibrinolysis.

Blood Coagul Fibrinolysis. 2004 Apr;15(3):213-9.

The authors compared the effect of 20ml per kg body weight crystalloid with 20 ml per kg bodyweight colloid. However, within that protocol the authors ignore that the volume effect of a crystalloid is at least four to five time lower and half life time of a crystalloid is at least 20 minutes compared to several hours compared to a synthetic colloid.

I would strongly recommend to renew and adapt the study protocol/methodology to some basic physiological principles.

Author Response

Reviewer #2

We would like to kindly thank you for reviewing our manuscript. Thank you for your comments and observations. Please find below our responses to your comments and suggestions and a summary of implemented revisions and corrections.

  1. Thank you for your suggestions regarding the past clinical studies researching the effects of colloids infusions. We acknowledged and referenced them in the „Discussion” section of our manuscript, along with some additional cited sources regarding the issue.
  2. Thank you for your comment regarding the expected potential physiological volume effects of colloid infusions. We did our best to improve in the „Discussion” section of our manuscript the rationale behind our study protocol and the methodology of blood samples collection. We included some additional references regarding the issue of vascular permeability and fluid volume effects in the setting of some additional confounding factors.

Reviewer 3 Report

The authors of the manuscript focused effects of balanced crystalloid or gelatine infusions to hemostasis. This influence of balanced crystalloid and synthetic gelatine infusions on coagulation and fibrinolysis in healthy volunteers included 25 males aged 18–30 years. Results were balanced crystalloid infusion provides less infusion-induced coagulopathy compared to gelatine.The authors have come up with an interesting article that is very important for medical research. Rotational thromboelastometry and thromboelastography is a holistic blood coagulation assay.  Rotational thromboelastometry is a viscoelastic hemostatic assay that has been used in emergencies (trauma and obstetrics), and surgical procedures (cardiac surgery and liver transplants), but experience with this assay in anticoagulant-treated patients is still limited. The positive thing about the manuscript is that the authors focus on its use of viscoelastic hemostatic assays. The manuscript is well structured, but some parts of the manuscript need to be corrected and supplemented.

Page 11, lines 284-287: MA/MCF in FIBTEM are important values that are critical in the use of fibrinogen concentrate in patients with fibrinogen disorder. This knowledge used in the perioperative management. We know about special clinical states for severe coagulopathies, where in perioperative management the patient must be given anticoagulation and coagulation treatment due to the high risk of thrombosis. Perioperative management of hypofibrinogenemic patients is complicated, requiring a specific assessment of the patient's overall hemostasis, taking into account both the bleeding and thrombotic risk. ROTEM can assist in the management, this was published in a manuscript that the authors should cite: Thromb Res. 2020 Apr;188:1-4. doi: 10.1016/j.thromres.2020.01.024. 

Page 11, lines 299-301: Clinical manifestations of bleeding are dependent on the fibrinogen levels. The bleeding phenotype usually develops in patients with fibrinogen levels lower than 1.0 g/L. A mean activity level of fibrinogen of at least 0.7 g/L prevents spontaneous hemorrhage. Authors should state these claims and cite the manuscript in which it was described: J. Clin. Med. 2022, 11(4), 1083; https://doi.org/10.3390/jcm11041083

Page 12, lines 325-331 I am missing a reference in this section. A review of the viscoelastic hemostatic assay was recently published. Authors should cite this manuscript: ,, J Clin Med. 2022 Feb 7;11(3):860. doi: 10.3390/jcm11030860“.

Tables and figures in the text are very clearly written.

I have to say that with these 42 references of which half of the references are from the last 5 years.

Author Response

Reviewer #3

We would like to thank you for your extensive review, substantive approach, and valuable comments and observations concerning the quality of our study. We acknowledge your kind comment regarding the quality of our investigation. We are glad you see the importance of the conducted research in the improvement of fluid therapy and fluid resuscitation management. We followed through with your suggestions. Please, find below the summary of implemented revisions and corrections.

  1. Page 11, lines 284-287: Thank you for this valuable piece of information. We deliberated upon the issue in the „Discussion” section of our manuscript and referenced the study you provided.
  2. Page 11, lines 299-301: This is vital knowledge in the clinical practice, so we would like to kindly thank you for this suggestion, and we acknowledged it as well in the „Discussion” section of the manuscript, with the cited source you supplied.
  3. Page 12, lines 325-331: Thank you for noticing this. We added the citation based on your reference.
  4. Thank you for acknowledging the quality of our tables and figures.
  5. Thank you for your observation regarding our cited sources. We did our best to provide up-to-date information based on the principles of Evidence-Based Medicine and high-quality studies.

Round 2

Reviewer 3 Report

The presented manuscript has been corrected in response to the suggestions. The authors have followed the recommendations of the reviewer. After the revision, the provided data and addition of the results became more clear. I would like to thank the authors for resubmitting the manuscript and explaining the obscure points from the previous version.